A deep learning approach for the detection and counting of colon cancer cells (HT-29 cells) bunches and impurities

Haq Inayatul 1
Mazhar Tehseen tehseenmazhar719@gmail.com 2
Naz Asif Rizwana rizwana@ncbae.edu.pk 3
Yasin Ghadi Yazeed 4
Saleem Rabea 5
Mallek Fatma 6
Hamam Habib 6 7 8 9 10
1 School of Electrical and Information Engineering, Zhengzhou University , Zhengzhou , Henan , China
2 Department of Computer Science, Virtual University of Pakistan , Lahore , Pakistan
3 School of Computer Science, National College of Business Administration and Economics , Lahore , Pakistan
4 Department of computer science and software engineering, Al Ain university , Abu Dhabi , United Arab Emirates
5 Department of computer science and software engineering, Air University , Multan , Pakistan
6 Faculty of Engineering, University of Moncton , Moncton , Canada
7 Spectrum of Knowledge Production, Skills Development , Sfax , Tunisia
8 College of Computer Science and Engineering, University of Ha’il , Ha’il , Saudi Arabia
9 International Institute of Technology and Management , Libreville , Commune d’Akanda , Gabon
10 Department of Electrical and Electronic Engineering Science, School of Electrical Engineering, University of Johannesburg , Johannesburg , South Africa
Wan Shibiao
Electronic publication date: 2023 Dec 5
Publication date: 2023
Volume: 9
Electronic Location ID: e1651
Received 2023 Jun 29; Accepted 2023 Sep 22
Copyright: ©2023 Haq et al.
Copyright year: 2023
Copyright holder: Haq et al.
License: This is an open access article distributed under the terms of the Creative Commons Attribution License, which permits unrestricted use, distribution, reproduction and adaptation in any medium and for any purpose provided that it is properly attributed. For attribution, the original author(s), title, publication source (PeerJ Computer Science) and either DOI or URL of the article must be cited.
License URL: https://creativecommons.org/licenses/by/4.0/

Keywords: Image processing, Deep learning, Lightweight network, Colon cancer cells, Colorectal, Adenocarcinoma, Cancer cells detection, IoMT, Neural networks, Healthcare

Funding: The authors received no funding for this work.

==============================
HT-29 has an epithelial appearance as a human colorectal cancer cell line. Early detection of colorectal cancer can enhance survival rates. This study aims to detect and count HT-29 cells using a deep-learning approach (ResNet-50). The cell lines were procured from Procell Life Science & Technology Co., Ltd. (Wuhan, China). Further, the dataset is self-prepared in lab experiments, cell culture, and collected 566 images. These images contain two classes; the HT-29 human colorectal adenocarcinoma cells (blue shapes in bunches) and impurities (tinny circular grey shapes). These images are annotated with the help of an image labeller as impurity and cancer cells. Then afterwards, the images are trained, validated, and tested against the deep learning approach ResNet50. Finally, in each image, the number of impurity and cancer cells are counted to find the accuracy of the proposed model. Accuracy and computational expense are used to gauge the network’s performance. Each model is tested ten times with a non-overlapping train and random test splits. The effect of data pre-processing is also examined and shown in several tasks. The results show an accuracy of 95.5% during training and 95.3% in validation for detecting and counting HT-29 cells. HT-29 cell detection and counting using deep learning is novel due to the scarcity of research in this area, the application of deep learning, and potential performance improvements over traditional methods. By addressing a gap in the literature, employing a unique dataset, and using custom model architecture, this approach contributes to advancing colon cancer understanding and diagnosis techniques.

Introduction

One of the most predominant malignancies and a major global cause of death is colon cancer. Significant advancements have been made in treating this prevalent illness in recent years. Adjuvant chemotherapy is successful, particularly for patients in stage III, and surgical techniques have been modified to provide the best results with the least amount of collateral harm. Many targeted medicines are currently in clinical trials, and some, like cetuximab and bevacizumab, have shown encouraging results, typically when administered in conjunction with chemotherapy. Only by developing current guidelines for use in clinical research and everyday practice can the most successful outcomes for these patients be attained (Labianca et al., 2010).

The colon, the largest segment of the large intestine, and the rectum, the shortest section, are possible sites for cancer growth. The colon is divided into four sections: the rectum, sigmoid colon, descending colon, and transverse colon. Polyps are growths that can form in the colon or rectum. These tumors are frequently the earliest sign of colorectal cancer. Detecting and eliminating polyps is a critical step in lowering the risk of colorectal cancer. Colorectal cancer causes cancerous cells to form in the colon or rectum. Screening and enhanced treatment options may have resulted in earlier disease discovery and more lives saved (Engstrom et al., 2009). HT-29 is a colorectal cancer cell line having an epithelial appearance. Colorectal cancer chemotherapy drugs 5′-fluorouracil and oxaliplatin can successfully eradicate these cells (Fogh, 2023). Figure 1 shows the structure and parts of the human colon (James et al., 2020).

Figure 1 Structure and parts of the human colon.

Diagnosing and treating colon cancer involves a multifaceted approach. Research using real cancer samples directly obtained from patients provides valuable insights into the disease’s specifics and treatment responses. However, practical challenges like ethical considerations and limited availability can hinder this. Research on established cancer cell lines bought from reputable companies is a common alternative. While not a perfect match for patient tumors, these cell lines offer controlled environments for studying cancer biology and therapies. Integrating findings from both approaches enriches our understanding of colon cancer and aids in developing effective diagnostic and treatment strategies (Zhou et al., 2020; Mirabelli, Coppola & Salvatore, 2019).

Hospitals utilise a range of techniques for detecting colon cancer cells. An essential approach is a colonoscopy, involving a tube with a camera to examine the colon lining and enable biopsies. Another method, flexible sigmoidoscopy, focuses on the lower colon. Stool tests such as FOBT, FIT, and DNA assess blood and genetic changes. Virtual colonoscopy deploys CT scans to identify issues non-invasively. Barium enemas use X-rays and barium liquid to visualise the colon’s shape—endoscopic ultrasound merges endoscopy and ultrasound to evaluate cancer spread. Blood tests track markers like CEA for progression (Baek et al., 2019).

The techniques for colon cancer detection have their limitations. Colonoscopy, while effective, can be uncomfortable and invasive, potentially leading to perforation and requiring sedation. Flexible sigmoidoscopy’s scope is limited, missing upper colon abnormalities and extensive biopsies. Stool tests may yield false results and demand preparation. Virtual colonoscopy exposes patients to radiation and struggles to differentiate polyp types. Double-contrast barium enema might overlook more minor issues. Endoscopic ultrasound’s quality depends on operators and only aids local staging. In blood test lack of specificity and variability presents challenges. Each technique has benefits, but its drawbacks emphasis the need for personalized approaches in colon cancer detection (Baek et al., 2019; Mauri et al., 2022).

AI techniques, particularly deep learning, can address the limitations of colon cancer detection methods. By enhancing image analysis, AI can improve colonoscopy and virtual colonoscopy, spotting subtle abnormalities and polyp detection. It enhances diagnostics, reducing errors in stool tests and aiding blood marker interpretation. AI enables personalized risk assessment, integrates diverse patient data for comprehensive insights, and lessens invasive procedures by refining virtual colonoscopies. Additionally, it offers decision support for healthcare professionals and enables continuous patient monitoring. However, data quality, interpretability, and regulation challenges need attention for successful integration (Huang et al., 2020; Lee & Yoon, 2021; Coccia, 2020). Figure 2 depicts the colon cancer (Mayo Clinic, 2023).

Figure 2 Colon cancer illustration.

ResNet is a deep neural network architecture designed to tackle the challenges of training deep networks. It uses skip connections to enable the training of intense models, making it practical for image classification tasks. While ResNet and similar deep learning models promise to detect cancer cells in medical images like colonoscopies accurately, their accuracy depends on data quality, model design, and task specifics. Ensembling, transfer learning, and collaboration with medical professionals are vital for accurate cancer cell detection in clinical settings (Duta et al., 2021; Gupta & Bibhu, 2023).

The approach outlined in this study involves obtaining a colon cancer cell line, generating an image dataset containing these cells, and utilising deep learning techniques for cell detection and counting. This process profoundly impacts cancer diagnosis and treatment research, contributing to researchers understanding of cancer behavior and potential treatments. By employing deep learning, authors advance disease modelling, gain insights into growth patterns and progression dynamics, and enhance early detection efforts. Additionally, the algorithm development holds promise for medical applications, aiding professionals in cancer diagnosis and monitoring. This work also extends to drug testing and treatment development, enabling the evaluation of potential therapies. Translational research potential arises from these findings, influencing diagnostic tools and treatment approaches. Incorporating deep learning amplifies impact, and the dedication to innovative research holds the potential to transform colon cancer management (Jain et al., 2023).

Contribution

The contribution of authors to this work is as follows.

• To prepare a dataset from the HT-29 cells line.

• To detect HT-29 cells and impurities using deep learning approach.

• To classify HT-29 cells and impurities using deep learning approach.

• To count colorectal cancer cells (HT-29 cells) and impurities using deep learning approaches.

Innovation and novelty

This work is essential in HT-29 cell detection and counting using DL. The scarcity of previous research in this area highlights the novelty of this approach. Utilising the powerful ResNet-50 deep learning model and curating a unique dataset in our lab, authors tackled a challenging problem. The ability to accurately detect and count HT-29 cells has crucial implications in various biomedical applications, such as cancer research, drug development, and disease modelling. This study has the potential to significantly contribute to the scientific community’s understanding of HT-29 cells and could pave the way for future advancements in the field. Table 1 shows the abbreviations and their meaning those are used in this study.

Table 1 The abbreviations used in this study.

Abbreviations	Full form	Abbreviations	Full form	
AI	Artificial intelligence	CAD	Computer-aided diagnostic system	
DL	Deep learning	LDCT	Low-dose computed tomography	
CT	Computerized tomography	I-ELCAP	International early lung cancer action project	
CEA	Carcinoembryonic antigen	SVM	Support vector machines	
ResNet	Residual network	DNN	Deep neural network	
DIP	Digital image processing	DenseNet	Densely connected convolutional neural network	
DLS	Deep learning system	BraTS	Brain tumor segmentation	
CNNs	Convolutional neural networks	FBS	Fetal bovine serum	
RNNs	Recurrent neural networks	DCNN	Deep convolutional neural network	
MLP	Multi-layer perceptron	GAP	Global average pooling	
ANN	Artificial neural network	ROI	Region of Interest	
SGD	Stochastic gradient descent	FN	False negative	
FPs	False positive	ML	Machine learning	

Related Works

DL and DIP methodologies were used to build a classification framework capable of examining histopathological images of lung and colon tissue and discriminating between five different types of tissue (two of benign and three of malignant). According to the statistics, the proposed framework can identify malignant tissues with a maximum accuracy of 96.33% (Masud et al., 2021).

Colon gland pictures were used to train ResNet-18 and ResNet-50 colorectal cancer classification models that distinguish between benign and malignant cases. According to the study, ResNet-50 surpasses ResNet-18 in three test data categories: accuracy, sensitivity, and specificity. The 20% and 25% test sets provide the best performance value, with classification accuracy over 80%, sensitivity exceeding 87%, and specificity exceeding 83% across all three test sets (Sarwinda et al., 2021).

The authors predicted disease-specific survival in stage II and stage III colorectal cancer using a DLS trained on 3652 cases. Annotators correctly detected this trait, which has a distinct visual appearance, with an accuracy range from 87.0% to 95.5% (Wulczyn et al., 2021).

CNNs and RNNs using whole-slide images from colon and stomach biopsy histopathology (RNNs). The models were trained to recognise non-neoplastic WSI, Adenocarcinoma, and adenoma. When the models were tested on three separate test sets, AUCs as high as 0.97 and 0.99 were achieved for gastric cancer and adenoma, respectively. In contrast, AUCs as high as 0.96 and 0.99 were obtained for colonic adenocarcinoma and adenoma, individually (Iizuka et al., 2020).

A DL-based cell classification and counting system were created using the inception (ResNet V2) feature extractor, a highly tuned architecture based on the FR-CNN. Compared to conventional Faster R-CNN, their technique increased the average precision of the tested data set from 71% to 85% (Albuquerque et al., 2021).

A CNN model was used to detect lung tumor in CT scan images. The CNN method was accurate 84.8% of the time, precise 90.4% of the time, and specific 90.64% of the time. These promising results indicate the algorithm’s potential to enhance lung cancer diagnosis by reducing false positives and aiding radiologists in identifying nodules accurately (Haq et al., 2022).

Supervised learning classifiers are employed to distinguish between normal and TB-infected lung images. The most accurate classifier, the MLP from the ANN, achieves 99% accuracy in less than a second. Other classifiers, including random forest, J48, Logit Boost, AdaBoostM1, and Bayes net, achieve high accuracy ranging from 96.83% to 98.83%. This approach demonstrates the potential for rapid and accurate TB detection using CT scan images (Haq et al., 2022).

The research employed a DL method to classify H&E stained breast tissue images into four categories: normal tissue, benign lesion, in situ carcinoma, and invasive carcinoma. Utilising a fine-tuned Inception-v3 CNN, the study prioritised patches with high nuclear density, especially in epithelial regions that display clear tumor indicators. The outcomes showed an accuracy of 85% for the four categories and 93% accuracy for the non-cancer class (Golatkar, Anand & Sethi, 2018).

A CAD was created using AlexNet to diagnose histopathological breast cancer images precisely. The study yielded remarkable performance metrics, including 95% accuracy, 97% sensitivity, and 90% specificity, illustrating the considerable potential of the proposed system for breast cancer diagnosis (Senan et al., 2021).

A CAD system is used for early lung nodule detection from LDCT images. The system’s effectiveness is assessed using 320 LDCT images from the I-ELCAP database. Notably, the study reveals that employing the VGG19 architecture alongside an SVM classifier yields the highest accuracy, achieving 96.25% for detection (Elnakib, Amer & Abou-Chadi, 2020).

A DNN was designed to detect lung cancer from CT images. The approach uses a DenseNet and an adaptive boosting algorithm to classify lung images into normal or malignant categories. The results of the tests showed that the suggested method was accurate 90.85% of the time (Kalaivani et al., 2020).

A modified U-Net structure was introduced based on residual networks for brain tumor segmentation. This modification incorporated periodic shuffling at the encoder section and sub-pixel convolution at the decoder section. The U-Net model’s performance was assessed using the BraTS challenge 2017 and 2018 benchmark datasets, resulting in segmentation accuracy of 93.4% and 92.2% (Pedada et al., 2023).

Gap in previous studies

The authors examined the previous studies and found a large volume of work on cancer detection, but there is a gap in detecting and counting colon cancer cells (HT-29 cells) using deep learning.

Methods and Techniques

This section explains a detailed methodology of “colon cancer cells” sample collection, Lab experiments, and a dataset of images containing colon cancer cells (HT-29 cells). The techniques for image processing (HT-29 cell detection, classification, and counting using DL algorithms) are also discussed.

Cell line and dataset collection

Cell culture

The cell lines were sourced from Procell Life Science & Technology Co., Ltd., in Wuhan, China. These cells were cultivated in McCoy’s 5A medium, enriched with 10% FBS, and maintained at 37 °C in an environment with 5% CO2 (Du et al., 2022).

Cells invasion and migration

For the assessment of cell migration and invasion, the authors of this study employed a transwell system comprising a 24-well transwell chamber (Jin et al., 2015). The experimental procedure involved several steps. Firstly, trypsin-treated cultured cells (1,104) were suspended in serum-free McCoy’s 5A medium and introduced into the upper chamber of the transwell. After a centrifugation step of 3 min at 800 g, 600 µl of McCoy’s 5A medium supplemented with 10% FBS was introduced into the lower chamber. The transwell chambers were pre-coated with a matrigel solution (1:4, BD, Franklin Lakes, NJ, USA) and incubated at 37 °C for 4 h to set up the invasion assay. Subsequently, the cells were allowed 24 h for migration in the experiment focusing on migration and 48 h in the experiment focusing on invasion. After the respective periods, cells that adhered to the filter’s upper side were carefully extracted. The cells stained with crystal violet and then attached to the membrane filter were enumerated using an Olympus microscope. This methodology effectively evaluated cell migration and invasion potential (Chen et al., 2016). Figure 3 illustrates the procedure of cell culture and dataset collection.

Figure 3 The procedure of dataset collection.

In Fig. 4, HT-29 cells are shown in blue shapes, while other shapes are impurities (Matrigel).

Figure 4 The blue shapes are HT-29 cells, and the others are impurities (Matrigel).

Proposed deep learning model

The proposed deep learning (DL) model is depicted in Fig. 5. With an Intel(R) Xeon(R) Bronze 3204 1.9 GHz processor and 62GB RAM, Linux is the operating system. The proposed architectures are trained and tested using the DL framework in Matlab 2021 for all experiments.

Figure 5 Proposed deep learning model.

ResNet-50 model architecture

For colon cancer, DL means teaching a model to look at medical pictures, like histopathological images of colon tissue samples, to help find and diagnose cancerous areas. A DCNN called ResNet-50 can be used as a feature extractor and classifier to find cancerous areas in these pictures.

In detecting and quantifying HT-29 cells in images, the ResNet-50 architecture is employed. The network’s input layer receives high-resolution images of Ht-29 cells, which have been examined under a microscope. These images can be in either black and white or color. The subsequent stages of the architecture involve convolutional and pooling layers designed to detect features of varying sizes within the images. By utilising a variety of convolutional layers with different filter sizes and implementing max-pooling operations, the architecture efficiently reduces the spatial dimensions. Central to ResNet-50’s structure are its residual blocks, encompassing 50 layers. These blocks incorporate skip connections, which aid in maintaining the smooth flow of gradients during training and facilitate the creation of deeper networks. While some residual blocks focus on learning localised image features, their overarching purpose is to enhance the network’s capacity to extract intricate patterns from the input images. GAP is introduced to distil a fixed-size feature vector for each image following the convolutional and pooling layers. This step consolidates the rich information extracted from the convolutional stages into a compact representation, preparing the data for further processing. In the architecture’s subsequent stages, one or more fully connected layers are employed to process the feature vector, yielding the final classification scores. These scores hold valuable information regarding the presence or absence of HT-29 cells within the input images. The ultimate output layer employs the softmax activation function to generate class probabilities, indicating the likelihood of a given tissue area being cancerous. This comprehensive ResNet-50 architecture effectively amalgamates these stages, enabling the precise detection and enumeration of HT-29 cells in images of colon tissue samples, thereby contributing to the advancement of cancer detection techniques (Sarwinda et al., 2021). Figure 6 depicts an architecture of ResNet-50 model (Mukherjee, 2022).

Figure 6 ResNet-50 model architecture (Mukherjee, 2022).

Data pre-processing and augmentation

DL models undoubtedly require significant data to avoid over-fitting issues. When the training data is more robust, the network’s generalization performance is at its best. The data augmented in this study make the network more robust to data volatility and ensure that the trained models can be deployed to various contexts and histopathology images. The augmentation methods are applied to the photos in the original dataset without increasing the number of shots. Each image is subjected to spatial and colour transformations. Each image and its associated binary mask are subjected to various spatial alteration methods. The photos are rotated vertically and horizontally at random and cropped with a zoom at random. Then, 90- and 180-degree rotations are used to imitate the various orientations from the pathologist’s point of view. Several preprocessing steps were employed to prepare the HT-29 images for training the ResNet-50 model. Initially, the images were scaled uniformly to dimensions of 224 ×224 pixels, aligning with the input size expected by ResNet-50.

Additionally, pixel values were scaled to a specific range, either [0, 1] or [−1, 1], ensuring consistent and manageable data representation. Data augmentation techniques enhanced the model’s generalization across different data variations. These techniques included random rotations, flips, and shifts, introducing controlled variations in the training dataset. Such augmentations aimed to expose the model to diverse perspectives of the HT29 cell images, facilitating better adaptability to varying scenarios.

Moreover, colour channel normalisation was executed to mitigate variations arising from differences in lighting conditions. This step homogenised the colour distribution across images, reducing the impact of lighting discrepancies during model training. By combining these preprocessing and augmentation strategies, the histopathological images of HT29 cells were effectively prepared for training the ResNet-50 model. This approach aimed to enhance the model’s robustness and accuracy when confronted with real-world data while maintaining the distinct characteristics of the HT29 cell images.

Hyperparameters

Several crucial hyperparameters are utilised to facilitate the training of a ResNet-50 model for HT-29 cell detection. The learning rate is a pivotal factor governing the adjustments made to weights in each training step, thereby influencing convergence speed. The batch size determines the number of images processed within each training iteration. The ResNet-50 architecture, which consists of 50 layers, is the foundational design employed. Standard optimisations such as Adam or SGD are implemented for optimisation during gradient descent. Weight decay is introduced to counteract overfitting as a regularisation mechanism.

Moreover, data augmentation techniques, encompassing random rotations, flips, and translations, are applied to enrich the dataset, fostering the broader adaptability of the model. Dropout layers are strategically incorporated to mitigate the risk of overfitting, causing specific neural units to be deactivated randomly during training. These hyperparameters collectively shape the training procedure, enabling the ResNet-50 model to detect HT-29 cells accurately.

Dataset labeling

The ROI Labeler tool in Matlab 2021 allows researchers to develop a variety of label-marking schemes interactively. In addition to scene labels, it can generate rectangle, polyline, pixel, and polygonal ROI labels in an image or picture sequence. The image labeller allows researchers to do the following:

• Load data without labels.

• Label an image frame by hand from a collection of images.

• Using an automation technique, label images automatically across frames.

• Export the data from the named ground truth.

• An ROI label represents a rectangle, polyline, pixel, or polygon region of interest. These labels contain the label name, such as “impurities”, and the production location.

• A scene label outline can outline the qualities of a scene. This label can be associated with a frame. Figure 7 shows the ROI and scene labeling.

Figure 8 shows the image annotation for impurities in green boxes. Figure 9 shows Image annotation as colorectal cancer cells (HT-29 cells) in the brown boxes.

Figure 7 ROI and scene labeling.

Figure 8 Image annotation as an impurity.

Figure 9 Image annotation as cancer cells (HT-29 cells).

The completed annotations of all classes (HT-29 cells and impurities) are shown in Fig. 10.

Figure 10 Image annotation as impurity and colorectal cancer cells (HT-29).

The accuracy graph of the ResNet50 DL approach is presented in Fig. 11. And Fig. 12 shows the Loss graph of the ResNet50 DL approach. Training is performed on these images. After training, the testing can detect and count the objects in the images.

Figure 11 Accuracy graph of ResNet50.

Figure 12 Loss graph of ResNet50.

Detection of HT-29 cells and impurities in the images after training are shown in Fig. 13.

Figure 13 Detection of HT-29 cells and impurities in images.

Counting of HT-29 cells and impurities

After training and testing, the cells will be counted for each image. An example of counting HT-29 cells and impurities is as follows.

>>sum(labels==’cancer’)

ans =

9

>>sum(labels==’impurities’)

ans =

8

As stated above, the same work has been done and counted the impurity and cancer cells in all images.

Classification of colon cancer histopathological images

All pictures were downsized to 224 × 224 × 3 patches to match the input size of each pre-trained model. A batch size of 32 ensures consistent feature representation throughout the learning process. Fifty training epochs were used for each model. SGD, a momentum-based optimiser, is used in this case. Initially, the learning rate was set to 0.0001. The learning rate is reduced by 0.1 every ten epochs to account for the data’s shifting hierarchical depths. If needed, a softmax-equipped added a final fully linked layer to identify patches consistently across diverse network configurations. This depth corresponds to the number of popular classes. Equation (4) allocates the class with the highest score to each patch based on the probabilistic representation produced by softmax, where yi represents the output of the final convolutional layer. Then the index of each output vector element is represented, and n indicates the number of classes. (1) Pyi=eyi ∑jneyj.

Performance evaluation

The following conventional measures are employed to assess and contrast proposed models with cutting-edge techniques:

(2) Accuracy=TP+TNTP+TN+FP+FN

(3) Miss Classification Rate=FP+FNTP+TN+FP+FN×100

(4) Specificity=TNTN+FP×100

(5) Sensitivity=Recall=TPTP+FN×100

(6) Precision=TPTP+FP×100

(7) Net Predictive value=TNTN+FN×100

(8) False Positive Rate=FPFP+TN×100

(9) False Discovery Rate=FPFP+TP×100

(10) False Omission Rate=FNFN+TN×100

(11) False Positive Ratio=1−Specificity100

(12) False Negative Ratio=1−Sensitivity100.

TN denotes the total number of true negatives, FN the total number of false negatives, FP the total number of false positives, and TP the total number of true positives.

Results and Discussion

This study presents 566 images of colon cancer and examines their results. All the images have cancer conditions and detection. The images are annotated with the help of an image labeller as impurity and cancer cells (HT-29 cells). Then, the images are trained, validated, and tested against the DL approach ResNet50. Finally, in each image, the number of impurity and cancer cells are counted to find the accuracy of the proposed DL model. Measurements for network performance, accuracy, and computational cost are used. Each model is trained and tested using ten independent non-overlapping trains and test random splits. Several tasks investigate and demonstrate the effects of data pre-processing. Table 2 shows the counting of HT-29 cells and impurities in images, and the total count and average values are shown in the last two rows.

Table 2 Counting of HT-29 cells and impurities in images.

Dataset	Colorectal Cancer Cells (HT-29)	Impurities	
Image. (1)	7	5	
Image. (2)	9	7	
Image. (3)	5	9	
Image. (4)	7	8	
Image. (5)	8	6	
Image. (6)	6	7	
Image. (7)	7	8	
Total	49	50	
Average	7	7.14	

The total counting procedure of colorectal cancer cells (HT-29) and impurities in images are presented in Table 3. The formula suggested for counting these classes in images is presented in the last column of the table.

Table 3 Total count of HT-29 and impurities in images.

Class	Total Number of Images	Counting	
Colorectal cancer cells (HT-29)	566	566 × 7 = 3962	
Impurity	566	566 × 7.14 = 4041.24	

The performance evaluation of the ResNet-50 DL approach is presented in Table 4. Here ReSNet-50 model has training and validation phases. Precision, F1 score, recall, and overall accuracy values are mentioned for HT-29 cells and impurities. A 95.5% accuracy during the training phase was obtained, and 95.3% for validation phase.

Table 4 Performance evaluation of the ResNet-50.

Model	Phase	Classes	Precision	F1 Score	Recall	Overall Accuracy	
ResNet-50	Training 70%	Impurity	0.96	0.95	0.93	95.50%	
		Cancer	0.97	0.96	0.96		
	Mean Value	0.965	0.955	0.945	
	Validation 30%	Impurity	0.95	0.94	0.93	95.30%	
		Cancer	0.97	0.96	0.97		
Mean Value	0.96	0.95	0.95	

The training phase of the ResNet-50 model is presented in Fig. 14, which contains the values of precision, F1 score, recall, and accuracies of HT-29 cells and impurities. The accuracy obtained for detecting and counting HT-29 cells and impurities during the training phase is 95.5%.

Figure 14 Training phase evaluation of ResNet-50 Model.

The validation Phase of the ResNet-50 model is presented in Fig. 15, which contains the values of precision, F1 score, recall, and accuracies of HT-29 cells and impurities. The accuracy obtained for detecting and counting HT-29 cells and impurities during the training phase is 95.3%.

Figure 15 Validation phase evaluation of ResNet-50 Model.

Comparison with some previous works

Searching in the previous literature shows limited studies regarding detecting and counting colon cancer cells (HT-29 cells). However, a lot of work has been done on the detection and classification of other cancer types. This work is compared with some previous relevant studies, as shown in the Table 5 and Fig. 16. Here the authors for the study of colorectal cancer using technique ResNet-50 is Sarwinda et al. (2021), and for DLS is Wulczyn et al. (2021). The authors for breast cancer using CNN and AlexNet are Golatkar, Anand & Sethi (2018) and Senan et al. (2021). Lung cancer study using techniques DenseNet and CNN are presented by Kalaivani et al. (2020) and Haq et al. (2022). The study using technique U-Net is presented by Pedada et al. (2023).

Table 5 Comparison with previous studies.

Authors	Cancer type	Technique	Accuracy	
Sarwinda et al. (2021)	Colorectal cancer	ResNet-50	80%	
Wulczyn et al. (2021)	Colorectal cancer	DLS	95.50%	
This study	HT-29 cells	ResNet-50	95.50%	
Golatkar, Anand & Sethi (2018)	Breast tumors	CNN	85%	
Senan et al. (2021)	Breast tumors	AlexNet	95%	
Kalaivani et al. (2020)	Lung tumors	DenseNet	90.85%	
Haq et al. (2022)	Lung tumors	CNN	84.80%	
Pedada et al. (2023)	Brain tumor	U-Net	93.40%	

Figure 16 Comparison of this work with other studies.

The Fig. 16 and Table 5 show that the proposed model best detects and counts congested and overlapping cells.

Conclusion

In this study, HT-29 cells and impurities are detected and counted using a deep learning technique (ResNet-50). Samples were collected from Procell Life Science & Technology Co., Ltd. for this purpose. The results of 566 images of colon cancer are presented and examined. All the images have cancer conditions for detection. These images were annotated with the help of an image labeller as impurity and cancer cells (HT-29 cells). Then the images are trained and validated against the DL approach ResNet-50. Finally, in each image, the number of impurity and cancer cells are counted to find the accuracy of the proposed ResNet-50 model. Accuracy and computational cost are measured as metrics for network performances. Ten separate non-overlapping trains and random test splits are used to train and test each model. Various challenges also studied and demonstrated the effects of data pre-processing. In the results, an accuracy of 95.5% was obtained during the training phase and 95.3% in the Validation phase for detecting and counting of HT-29 cells and impurities.

limitations of this Study

The utilisation of DL and the ResNet-50 model for HT-29 cell detection and counting holds promising implications for disease diagnosis and treatment. However, it is crucial to acknowledge certain limitations to avoid potential misinterpretations. Firstly, the model’s generalisation to other cell types might be limited, necessitating validation on diverse cell lines. Secondly, the accuracy heavily relies on the quality and representativeness of the dataset used for training, as biased or insufficiently diverse samples can affect its performance. Moreover, variations in tissue preparation, staining, and imaging conditions can introduce artefacts impacting the model’s accuracy. FPs and FNs also need consideration, along with the importance of clinical validation for medical applications. Addressing regulatory and ethical aspects is imperative, ensuring compliance with guidelines and maintaining transparency and data privacy standards in implementing DL models for disease diagnosis and treatment. Annotating congested and overlapped cells is challenging for non-healthcare persons.

Future Work

This research assigned task was to detect and count colorectal cancer cells (HT29 cells) using a DL model (ResNet-50). However, in the future, authors aim to classify, identify and count different cells like; RBCs, WBCs, etc., along with cancer cells in every image using DL (multiple objects detection) algorithms. Researchers can also detect cancer cells in different body areas like Lungs, breasts, brain, and Liver, etc. In future studies, authors will focus on improving the accuracy with more variants of ML and DL approaches to get high accuracy. Furthermore, the federated learning approach will be applied to centralise and secure the data or information to promote this technology to the clinic as soon as possible.

Supplemental Information

Supplemental Information 1 Code

Click here for additional data file.

Supplemental Information 2 HT-29 Cells Dataset 1: Image series 0+1 to 2-14

Click here for additional data file.

Supplemental Information 3 HT-29 Cells Dataset 2: Image series 4+9 to 5-17

Click here for additional data file.

Supplemental Information 4 HT-29 Cells Dataset 3: Image series 2-15 to 4+8

Click here for additional data file.

Supplemental Information 5 HT-29 Cells Dataset 4: Image series 0+1 to 2+1

Click here for additional data file.

Supplemental Information 6 HT-29 Cells Dataset 5: Image series 2+2 to 10+6

Click here for additional data file.

Supplemental Information 7 HT-29 Cells Dataset 6: Image series 0+1 to 2-3

Click here for additional data file.

Supplemental Information 8 HT-29 Cells Dataset 7: Image series 2-4 to 4-12

Click here for additional data file.

Supplemental Information 9 HT-29 Cells Dataset 8: Image series 1178 to 1243

Click here for additional data file.

Supplemental Information 10 HT-29 Cells Dataset 9: Image series 0+1 to 2+8

Click here for additional data file.

Supplemental Information 11 HT-29 Cells Dataset 10: Image series 2+9 to 5-15

Click here for additional data file.

Supplemental Information 12 HT-29 Cells Dataset 11: Image series 4+4 to 5-14

Click here for additional data file.

Additional Information and Declarations

Competing Interests

Author Contributions

Data Availability

The authors declare there are no competing interests.

Inayatul Haq conceived and designed the experiments, performed the experiments, analyzed the data, performed the computation work, prepared figures and/or tables, authored or reviewed drafts of the article, data collection, Proofreading, and approved the final draft.

Tehseen Mazhar conceived and designed the experiments, performed the experiments, analyzed the data, performed the computation work, prepared figures and/or tables, authored or reviewed drafts of the article, writing draft, Formal Analysis, validation, and approved the final draft.

Rizwana Naz Asif conceived and designed the experiments, performed the experiments, analyzed the data, performed the computation work, prepared figures and/or tables, authored or reviewed drafts of the article, methods, investigation, and approved the final draft.

Yazeed Yasin Ghadi conceived and designed the experiments, performed the experiments, analyzed the data, performed the computation work, prepared figures and/or tables, authored or reviewed drafts of the article, conceptualization, investigation, and approved the final draft.

Rabea Saleem conceived and designed the experiments, performed the experiments, analyzed the data, performed the computation work, prepared figures and/or tables, authored or reviewed drafts of the article, review and editing, and approved the final draft.

Fatma Mallek conceived and designed the experiments, performed the experiments, analyzed the data, performed the computation work, prepared figures and/or tables, authored or reviewed drafts of the article, project administration, resources, and approved the final draft.

Habib Hamam conceived and designed the experiments, performed the experiments, analyzed the data, performed the computation work, prepared figures and/or tables, authored or reviewed drafts of the article, funding acquisition, supervision, and approved the final draft.

The following information was supplied regarding data availability:

The dataset and code are available in the Supplemental Files.

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
