# Peer review of "A deep learning approach for the detection and counting of colon cancer cells (HT-29 cells) bunches and impurities"

_PeerJ Computer Science, doi:10.7717/peerj-cs.1651_

## Round 0.1 · original submission · Major Revisions

The reviewers have substantial concerns about this manuscript. The authors should provide point-to-point responses to address all the concerns and provide a revised manuscript with the revised parts being marked in different color.

**Language Note:** The review process has identified that the English language must be improved. PeerJ can provide language editing services - please contact us at copyediting@peerj.com for pricing (be sure to provide your manuscript number and title). Alternatively, you should make your own arrangements to improve the language quality and provide details in your response letter. – PeerJ Staff

Reviewer 1 ·

Basic reporting

The paper is very well-written, well-structured and provides detailed descriptions of the proposed methodology and its evaluation

Experimental design

The experimental design and data are enough to explain the objective of the paper.

Validity of the findings

The innovativeness is moderate.

Additional comments

In this manuscript, the authors used the ResNet-50 model to detect and count colon cancer cells and impurities. On this basis, they separated image databases into two classes, labeled the images, and tested them using a deep-learning approach. This work provides new insight and opinion into detecting cells and impurities. The manuscript is well-organized and clearly stated. I would suggest accepting it after the following minor concerns are addressed.
1.What are the spatial alteration methods used on each image and its associated binary mask?
2.Can you describe Figure 13 and Figure 14 and their significance in relation to the ResNet-50 Model?

Reviewer 2 ·

Basic reporting

General Comments:
The manuscript presents a deep learning method for the detection and counting of a cultured cell line, specifically colon cancer cells and impurities. However, the major concern regarding this paper is that the proposed method is not based on real circulating cancer cells or patient biopsy immunohistochemistry (IHC), and thus, its relevance to disease diagnosis or treatment is limited. Furthermore, the manuscript suffers from numerous issues, including redundant content, grammar and spelling mistakes, lack of explanation regarding the term "impurities," missing comparisons with other studies, overuse of first-person pronouns, and inconsistent formatting. Additionally, the figures are of low quality, with issues such as low resolution, small fonts making them difficult to read, and errors in figure labeling.
Specific Comments:
Relevance and Contribution:
The main concern with this paper is the lack of relevance to disease diagnosis or treatment. While the deep learning method for detecting and counting colon cancer cells and impurities is of interest for cell culture research, it does not directly address real circulating cancer cells or patient biopsy IHC. The authors should clearly state the limitations of their method in the context of disease diagnosis and treatment to avoid any potential misinterpretation.
Content and Writing Style:
The manuscript contains redundant writing, presenting content that is not directly related to the core study. The authors should streamline the text and focus on essential aspects related to the deep learning method and its application to colon cancer cell detection and counting. Unnecessary content should be removed or placed in supplementary material, if applicable.
1. Here are some lines that the author may consider to shorten or omit: Line 71-155.
2. Line 71: “Remember that these receptors are…” The authors did not mention anything about what the receptors are and why they are important.
Clarity and Explanation:
1. The manuscript lacks a clear explanation of what the term "impurities" refers to. The authors must define and describe these impurities explicitly, as they play a crucial role in the study.
2. Additionally, the authors should provide a comprehensive explanation of the advantages of their study compared to existing methods or approaches in the field. For example: Line 177-179, what is the efficiency of CTC (circulating tumor cells) detection using the algorithm in this study? (It seems that the authors did not do CTC).
3. Line 180-181: the authors need to explain what is the advantage of using MATLAB over Python.
Language and Formatting:
The manuscript contains several grammar and spelling mistakes, which negatively impact its overall readability. The authors should carefully proofread the text to correct these errors. Furthermore, the inconsistent use of different fonts, capital letters, and non-capital letters is distracting and unprofessional. The entire manuscript should adhere to a consistent formatting style to improve its presentation. The following are just some examples:
1. Line 75, a kind of Cancer that  a kind of cancer that
2. Line 96, Adenocarcinoma
3. Line 98, Cancer, Colon
4. Line 191, Sructure  Structure
5. Mix fonts: Line 462, 474, 495, 509
6. Many others…
Use of Pronouns:
The manuscript excessively uses first-person pronouns ("we," "our"), which can affect the objectivity and overall tone of the paper. The authors should revise the text to maintain a more neutral and objective writing style.
Figure Quality:
The quality of figures, especially Figure 5, is inadequate. The low resolution and small fonts used in the figures make it difficult for readers to read. Moreover, the mislabeling of Figure 6, where the brown-colored label indicates HT-29 cells instead of impurity, is a significant issue that needs immediate correction. All figures should be of high resolution, with clear and legible labels.

Experimental design

The main concern with this paper is the lack of relevance to disease diagnosis or treatment. While the deep learning method for detecting and counting colon cancer cells and impurities is of interest for cell culture research, it does not directly address real circulating cancer cells or patient biopsy IHC. The authors should clearly state the limitations of their method in the context of disease diagnosis and treatment to avoid any potential misinterpretation.

Validity of the findings

Samples including circulating tumor cells and biopsy IHC samples are not included to validate the deep learning method.

Additional comments

Given the major concern regarding the relevance of the proposed method to disease diagnosis or treatment, the numerous issues related to content, language, and figure quality, it is my recommendation to make significant revisions on this manuscript. If the authors can address the concerns raised and make substantial improvements to the manuscript, including the relevance of the proposed method, clarity of explanations, and overall writing quality, they may consider resubmitting it for further review.

Annotated reviews are not available for download in order to protect the identity of reviewers who chose to remain anonymous.

Reviewer 3 ·

Basic reporting

The paper describes the use of the ResNet-50 model for detection and counting. While this is a widely used deep learning approach, additional details about the model's hyperparameters, training process, and optimization techniques would improve the reproducibility and understanding of the study. The paper isn’t well organized. To meet the increasingly high-quality standard of the journal, I have some points below.

1. The layout of the article is a bit messy, and I hope it can be improved.

2. The author mentions the DLC method both in the abstract and in the text, but doesn't seem to explain what the acronym is, nor does he describe the method in detail.

3. It lacks sufficient detail. The paper should provide a comprehensive description of the deep learning approach, ResNet-50 model's architecture, hyperparameters, data pre-processing techniques, and the evaluation process.

4. The author thinks that it is a novelty for others to use python while using MATLAB to implement the algorithm, which is inappropriate.

Experimental design

no comment

Validity of the findings

no comment

Additional comments

no comment

---

## Round 0.2 · accepted · Accept

The authors have addressed the concerns raised by the reviewers. I suggest accepting this manuscript.

Reviewer 3 ·

Basic reporting

I appreciate authors’ efforts in conducting response to deal with my questions. The authors’ intentions are correct, and their efforts should be encouraged. The authors have basically addressed the points raised by this Reviewer in the revised version.


Recommendation: After formatted, considered for Acceptation.

Experimental design

no comment

Validity of the findings

no comment